# Automotive 2.1 μm Full-Depth Deep Trench Isolation CMOS Image Sensor with a 120 dB Single-Exposure Dynamic Range [note 1]

**DOI:** 10.3390/s23229150

**Published:** 2023-11-13

**Authors:** Dongsuk Yoo, Youngtae Jang, Youngchan Kim, Jihun Shin, Kangsun Lee, Seok-Yong Park, Seungho Shin, Hongsuk Lee, Seojoo Kim, Joongseok Park, Cheonho Park, Moosup Lim, Hyungjin Bae, Soeun Park, Minwook Jung, Sungkwan Kim, Shinyeol Choi, Sejun Kim, Jinkyeong Heo, Hojoon Lee, KyungChoon Lee, Youngkyun Jeong, Youngsun Oh, Min-Sun Keel, Bumsuk Kim, Haechang Lee, JungChak Ahn

**Affiliations:** Samsung Electronics Co., Ltd., Yongin-si 17113, Republic of Korea; ds.yoo@samsung.com (D.Y.); yt.jang@samsung.com (Y.J.); yc1306.kim@samsung.com (Y.K.); hongsuk2.lee@samsung.com (H.L.); seojoo.kim@samsung.com (S.K.); moosup.lim@samsung.com (M.L.); hyungjin.bae@samsung.com (H.B.); hchang.lee@samsung.com (H.L.); jungchak.ahn@samsung.com (J.A.)

**Keywords:** automotive, HDR, LFM, sub-pixel structure, F-DTI

## Abstract

An automotive 2.1 μm CMOS image sensor has been developed with a full-depth deep trench isolation and an advanced readout circuit technology. To achieve a high dynamic range, we employ a sub-pixel structure featuring a high conversion gain of a large photodiode and a lateral overflow of a small photodiode connected to an in-pixel storage capacitor. With the sensitivity ratio of 10, the expanded dynamic range could reach 120 dB at 85 °C by realizing a low random noise of 0.83 e- and a high overflow capacity of 210 ke-. An over 25 dB signal-to-noise ratio is achieved during HDR image synthesis by increasing the full-well capacity of the small photodiode up to 10,000 e- and suppressing the floating diffusion leakage current at 105 °C.

## 1. Introduction

High dynamic range (HDR) is a primary requirement for automotive image sensors to capture not only both bright and dark regions in a single scene, but also the light-emitting diodes’ (LEDs) lighting signals of transportation environments. One way to expand the dynamic range (DR) is to use the multiple exposure method [1,2]. However, the sequential image capture causes motion artifacts and the short integration time results in LED flickering. Thus, a single exposure with a longer integration time is necessary along with HDR. To satisfy these, the lateral overflow integration capacitor (LOFIC) technology has been introduced, which can store excess electrons generated during the long exposure time using a large in-pixel capacitor [3,4,5,6]. The high-capacity MOS and metal–insulator–metal capacitors can be used as in-pixel capacitors, of which the unit capacitance of the in-pixel capacitors is known to be around few 10 fF/μm^2^, corresponding to about 60 ke-/μm^2^ [1]. The single photodiode (PD) with LOFIC has achieved a single-exposure DR of 110 dB [5], whereas LOFIC combined with a sub-pixel structure can expand the DR up to 120 dB in a single exposure [6]. The spatial sampling with a high sensitivity ratio between a single large PD (LPD) and a single small PD (SPD) enables it to expand the DR.

Along with HDR, automotive sensor technology using the sub-pixel structure also requires the improvement of sensing and viewing image quality at high temperature. However, when synthesizing the low light and bright images captured from the two PDs, signal-to-noise ratio (SNR) degradation occurs and the SNR is further deteriorated at high temperature. The recent work has reported that reducing sensor noises at high temperature can prevent the large SNR degradation in HDR images [6]. In addition, for a high-resolution HDR image, the technology cannot evade pixel shrinking. However, as the pixel size gradually shrinks, this sub-pixel architecture not only faces a critical limit where the DR is no longer expanded due to the restricted pixel area for the in-pixel capacitor, but also noticeably deteriorates SNR during HDR image synthesis. The high-density in-pixel capacitor approach could overcome the former restriction and the technology is still evolving through a variety of the high capacity capacitors [5,7]. However, such scaling inherently creates SNR degradation as a result of the small full-well capacity (FWC) of the PDs.

Despite the restrictions, here, we have developed a 2.1 μm CMOS image sensor incorporated with a sub-pixel structure by applying the full-depth deep trench isolation (F-DTI) process [8]. A single-exposure DR of 120 dB is achieved at a junction temperature (T_j_) of 85 °C by using a storage capacitor in a pixel. By increasing the FWC of the SPD and reducing the floating diffusion (FD) leakage current, during the HDR image synthesis, the minimum SNR is improved up to 25 dB at T_j_ 105 °C. We show that integrating the F-DTI process into a pixel array can increase FWCs while reducing the size of both LPD and SPD with a competitive HDR performance in the automotive image sensor technology.

## 2. Pixel Design and Operation

### 2.1. A 2.1 μm F-DTI Pixel Capped with a Storage Capacitor

As the pixel size shrinks for high resolution, in the sub-pixel structure, the size of both LPD and SPD continues to decrease. In particular, the size of the SPD becomes less than 1 μm and the conventional back-side deep trench isolation (B-DTI) process is restrictive due to the limitation of forming potential barriers between the PDs. B-DTI pixels less than 1 μm do not exceed the FWC of 7000 e- or more unless a deeper PD is formed [9]. Therefore, the larger FWC of the SPD with no image lag may not be feasible; it is difficult to scale down with the B-DTI pixel structure. To overcome the limitation, we utilized a F-DTI process in the way of maximizing the FWC of both PDs while suppressing optical and electrical crosstalk and blooming. A 2.1 μm F-DTI sub-pixel structure is shown in Figure 1.

A physically isolated sub-pixel structure is fabricated, as shown in Figure 1a. A fully depleted region of the F-DTI pixel can be extended to the deep-level of the silicon via high-energy n-type implantation and a large FWC of an SPD is demonstrated via the device simulation, as shown in Figure 1b. On the front side of the silicon surface, the transistors and FDs are separated by an additional p-well implantation, further enlarging the FWC of the buried PDs. The scanning electron microscope (SEM) images of the cross-sectional and top views for the 2.1 μm F-DTI sub-pixel structure are shown in Figure 1c. With the vertical transfer gate (VTG) and the buried PDs, an FWC of the SPD as high as 10,000 e- is realized in order to improve SNR. 

By introducing a storage capacitor within the pixel, the F-DTI pixel capped with the capacitor allows us to expand the DR with a lateral overflow operation. The capacitance of the overflow capacitor is more than 34 fF per pixel, allowing more than 210 ke- to be accumulated in a single exposure. Micro-lens and grids of LPD and SPD are also optimized to have a sensitivity ratio of 10 to further expand the DR with the merit of the sub-pixel structure.

### 2.2. Pixel Circuit and Operation

Figure 2a shows the pixel schematic of the 2.1 μm F-DTI pixel, consisting of two PDs, three FDs, eight transistors, and one storage capacitor. Using a DRG transistor, the LPD supports a dual conversion gain (CG) readout, and the SPD with a low CG due to the capacitor can be switched via an SW transistor. When all transistors are turned on, every FD node is connected to the gate of a source follower (SF) amplifier and then the circuit can operate in the readout mode of LOFIC signals. To reduce the discharge time of the storage capacitor, a DSW transistor is introduced. The left and right VDDA are separated and a reset voltage of the capacitor becomes the right VDDA to provide a stable supply voltage during the reset operation via the DSW transistor. We implement a four-readout scheme: correlated double sampling (CDS) and incomplete CDS for both LPD and SPD. A simplified timing diagram is shown in Figure 2b. During the operation, the sequential readouts are performed with LPD-HCG (LPD with high CG readout), LPD-LCG (LPD with low CG readout), SPD-CDS (SPD with CDS readout), and SPD-LOF (SPD with LOFIC readout). CDS is applied to LPD-HCG and SPD-CDS, while incomplete CDS is applied to LPD-LCG and SPD-LOF.

Initially, LPD is reset via the shutter operation and photoelectrons are generated in LPD during an integration time of 11 ms. After the integration time, RG and DRG turn off and a reset level is sampled to FD1. After that, when LTG is turned on, the electrons are transferred to FD1 and a signal level is sampled, so a complete CDS operation is performed for LPD-HCG. More photoelectrons can be stored in the additional capacitance of FD2 by turning DRG on. The LPD-LCG with an incomplete CDS readout starts after the LPD-HCG by sampling a signal level. To apply CDS to both HCG and LCG signals in the LPD, additional sampling capacitors and comparator circuits are required to store the LCG reset voltage [6]. Without a spatial cost, we adopt a single readout circuit for the complete CDS of HCG and the incomplete CDS of LCG. Side effects from the incomplete CDS, such as a power supply reject ratio (PSRR), will be mitigated using PSRR compensation circuit techniques. Since the reset readout is performed after the signal readout during an incomplete CDS operation, the power supply noise has no correlation between the reset and signal readouts. Thus, the incomplete CDS readout has a lower PSRR than complete CDS. To improve the power-supply noise immunity, internal voltage regulators are implemented and a PSRR compensation circuit is used, as shown in Figure 3a. The power noise of pixels at the pixel output node INP are subtracted during the comparator operation by the ramp buffer output node INN, which includes the power noise injected at the PSRR compensation circuit. The amount of the injected power noise is determined using the ratio of capacitors connected to the power and the ground. The improved PSRR performance is shown in Figure 3b.

After the LPD readouts, FD3, FD2, and FD1 are successively connected together by turning on SW and DRG, and the voltage level of FD1 becomes a reset level of SPD-CDS. After STG is turned on, photoelectrons from SPD are transferred and sampled to FD1, so that the complete CDS is performed for SPD-CDS. Continuously, the signal level of the overflow photoelectrons generated at high illuminance are firstly read out as they have already accumulated in the in-pixel storage capacitor. After DRG is turned off and SW and RG are turned on, the in-pixel capacitor is discharged to the reset level. After turning DRG on again, the reset level is sampled at FD1 by compensating the clock feedthrough voltage level of DRG from the low-to-high transition. The operation cancels the feedthrough voltage level of the DRG transition during the SPD-CDS readout, which is included at the signal level of SPD-LOF. Consequently, this operation becomes the incomplete CDS of the SPD-LOF.

Since a large number of photoelectrons can be stored in the in-pixel capacitor, the addition of the DSW transistor enables fast discharge by shortening the discharge path of the capacitor. In Figure 4a, the simulation shows that the reset settling time of 1 µs is achieved by adding DSW, to support the higher frame rate. The simulated results are verified using the measurements, as shown in Figure 4b. The reset settling has been measured; the average output code of SPD-LOF is set to 2000 LSB; and if reset sampling is incomplete, the output code will be less than 2000 LSB. The result confirms that we have achieved the settling time of less than 1 µs by applying the DSW transistor.

## 3. Results and Discussion

### 3.1. HDR Characteristics

Based on the *DR* defined as the ratio between the exposure level at 0-dB (*SNR*1) and the saturation exposure level, one can express a single-exposure *DR* of an equation as below.
(1)$$DR\left(dB\right)=20\times log(\;\frac{Sensitivity\;ratio\times Overflow\;capacity}{{Noise}_{@SNR1}}\;)$$

With the LOFIC operation, the single-exposure *DR* of the sub-pixel structure can be expanded by increasing the sensitivity ratio and the capacity of the storage capacitor, as well as reducing the total noises at SNR1. At high temperature, the dark random noise (RN) and dark signal non-uniformity (DSNU) of LPD-HCG are expected to be one of the major factors that lower the DR. To reduce RN, the width and length of the SF were optimized and the high CG of 185 μV/e- was achieved by reducing the FD capacitance of the LPD. As a result, the input-referred RN is measured to be 0.83 e-, enabling DR expansion by achieving the low SNR1. Additionally, the sub-pixel structure consists of the high sensitivity of an LPD and the low sensitivity of an SPD, covering a wide range of illuminance from dark to bright environments. Since an in-pixel storage capacitor is connected to the low sensitivity of SPD, the DR can be enormously expanded. By the aid of the high sensitivity ratio, a single-exposure DR of 120 dB was obtained at T_j_ 85 °C.

The photoelectrons, captured from LPD and SPD, are merged to synthesize the HDR image. However, due to the low sensitivity of SPD, a large SNR drop has been found during the LPD-to-SPD transition. As the sensitivity ratio increases, the DR is expanded, but the SNR dip becomes even lower. This SNR degradation can be prevented by introducing the SPD-CDS readout mode [6]. When the SPD-CDS readout (gray dashed line) is implemented, the second SNR dip is greatly improved up to 25 dB at 105 °C, as shown in Figure 5a. The SNR dip can be further improved by increasing the FWC of LPD to improve SNR or reducing the RN noise of SPD-CDS by introducing a high conversion gain of the SPD-CDS mode [7]. When LOFIC operates, a third SNR dip becomes rather prominent and is determined by a noise floor of the incomplete CDS of SPD-LOF. The temperature-dependent noises, such as thermal noise (kTC noise) and FD contact leakage, additionally remain, and the SNR dip is expected to be further deteriorated at high temperature. We performed two experiments to prevent the SNR drop: increasing the FWC of SPD and decreasing the noise floor of SPD-LOF.

In Figure 5a, the estimated SNR curve of the 2.1 μm F-DTI pixel is compared with the SPD FWCs of 5000 e- and 10,000 e- at T_j_ 105 °C. By maximizing the FWC of SPD up to 10,000 e-, the SNR dip is improved by more than 27 dB. As the temperature increases, however, the third SNR dip gets worse, becoming a minimum SNR dip over the entire DR. At high temperature, the DSNU noise becomes a dominant noise floor of SPD-LOF, which ultimately determines the SNR level. Since the main source of the DSNU is the dark current generation in the FD contact (FD3), DSNU tends to increase as the FD leakage current increases. The FD contact leakage is exponentially increased as the temperature increases, and a similar behavior has been found in DSNU, as shown in Figure 6. We adopt an additional surface curing process, where the FD contact leakage and the DSNU of the 2.1 μm F-DTI pixel exhibits a great reduction. With the defect curing process, we finally achieve the third SNR dip of approximately 30 dB at T_j_ 105 °C, as shown in Figure 5b.

### 3.2. Sensor Characteristics

The 2.1 μm F-DTI pixel is fabricated in a 65 nm stacked BSI process and the sensor performance is summarized in Table 1. The peripheral circuit of the sensor consists of a 3840 H × 2160 V pixel array, column analog-to-digital converter array, and an on-chip timing generator. The FWC of SPD reaches 10,000 e- for SNR improvement and the high capacity of the storage capacitor is designed to be more than 34 fF, enabling us to store 210 ke-/per pixel for DR expansion. Thanks to the merits of the F-DTI process, the single-exposure DR of 120 dB is achieved with the SNR dip greater than 25 dB at high temperature.

Along with the very competitive HDR characteristics, our 2.1 μm F-DTI pixel supports various color filter arrays: RCCB, RCCG, RGGB, and RYYCy. The relative QE of RCCB color filters shows a similar spectral response between LPD and SPD, as shown in Figure 7. The similarity of the spectral response is important to suppress the color noise during the HDR image synthesis with linearity improvement.

A micrograph of the fabricated test chip is shown in Figure 8. The chip was assembled in ball grid array (BGA) packaging, where the net die is connected to wire bonds and encapsulated with a transparent molding material. Using the test chip, an HDR sample image is captured inside a tunnel, as shown in Figure 9. With the HDR and SNR improvement, the objects inside and outside the tunnel are clearly recognized. The single-exposure DR of 120 dB shows that the outside the tunnel is no longer saturated even in the bright sunlight.

With the integration time of 11 ms, additional HDR images are taken with LED traffic lights under outdoor sunlight and fast-moving objects in the dark outside, as shown in Figure 10. Without an HDR mode, the LED traffic light cannot be detected, while the LED flicker is mitigated by using the HDR mode. The HDR of the 120 dB with the long integration time is sufficient to capture LED pulses of the traffic light without signal loss. In addition, with the HDR mode, the motion artifacts of the motorcycle LED headlight can be reduced. With the improvement of sensing and viewing capabilities, the 2.1 μm F-DTI pixel can be applied to both an advanced driver-assistance system (ADAS) and viewing sensors for a vehicle. Table 2 summarizes the representative HDR features compared to the previous works.

## 4. Conclusions

Using the F-DTI process, we have developed the 2.1 μm F-DTI pixel with LED flicker mitigation for automotive applications. The single-exposure DR of 120 dB is achieved by employing the sub-pixel architecture capped with an in-pixel storage capacitor. The capacitor can enormously extend the charge-accumulating capability of the pixel with a fast readout operation, supported by the additional DSW transistor. During HDR image synthesis, the minimum SNR degradation can be improved over 25 dB at high temperature. With the aids of both F-DTI and surface curing processes, the resulting improvement in SNR is attributed to the increase in FWC of SPD and the decrease in DSNU of SPD-LOF. Through the design approach with the F-DTI process, we believe that it is possible to accelerate the development of automotive sensor technology with superior HDR characteristics as well as image quality.

## Figures and Tables

**Figure 1 sensors-23-09150-f001:**
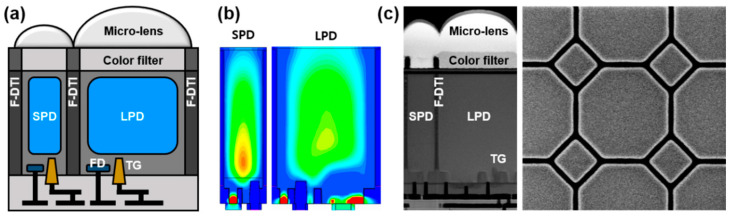
Cross-sectional views of (**a**) a sub-pixel structure and (**b**) simulated electrostatic potential profiles, and (**c**) cross-sectional and top-view SEM images of a 2.1 μm F-DTI pixel.

**Figure 2 sensors-23-09150-f002:**
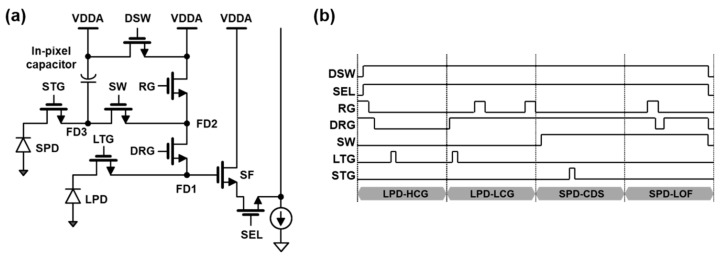
The proposed pixel circuit with readout timing: (**a**) a pixel schematic of the 2.1 μm F-DTI pixel with an in-pixel storage capacitor and (**b**) a four-readout timing diagram.

**Figure 3 sensors-23-09150-f003:**
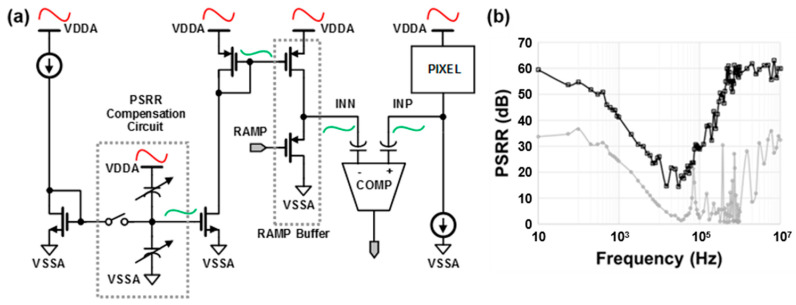
The concept scheme of (**a**) PSRR compensation circuit and (**b**) the measured results before (gray) and after (black) applying PSRR compensation.

**Figure 4 sensors-23-09150-f004:**
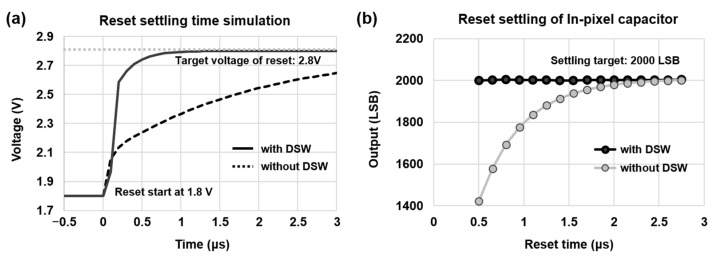
The comparison of storage capacitor reset settling: (**a**) simulated results of the reset settling at FD3 in SPD and (**b**) the measured results of SPD-LOF output signals with and without DSW.

**Figure 5 sensors-23-09150-f005:**
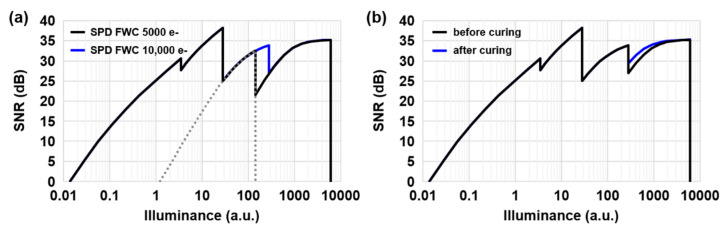
The comparison of SNR curves: (**a**) SPD of 5000 e- and 10,000 e- FWCs in 2.1 μm F-DTI pixels and (**b**) the noise effects before and after defect curing process. From left to right, the SNR curves of LPD-HCG, LPD-LCG, SPD-CDS, and SPD-LOF are estimated using the measured parameters, including total noises (temporal and spatial noises) during 11 ms integration time at T_j_ 105 °C.

**Figure 6 sensors-23-09150-f006:**
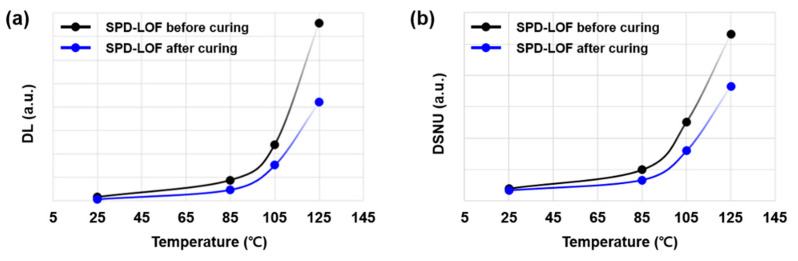
Temperature dependent noise floors: (**a**) dark current level (DL) of FD contact leakage and (**b**) DSNU of SPD-LOF during 11 ms integration time. The black and blue dot lines indicate before and after adopting the surface curing process, respectively.

**Figure 7 sensors-23-09150-f007:**
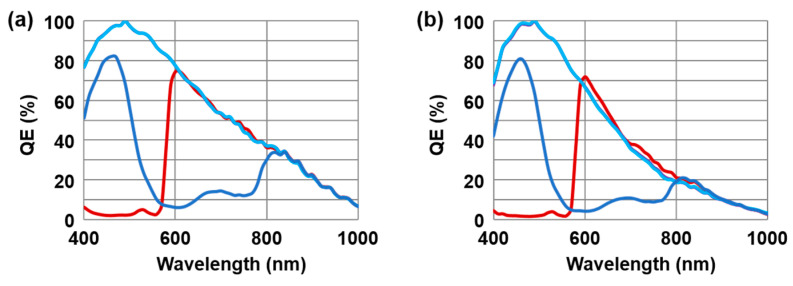
Relative quantum efficiency (QE) of (**a**) LPD and (**b**) SPD of RCCB color filter array.

**Figure 8 sensors-23-09150-f008:**
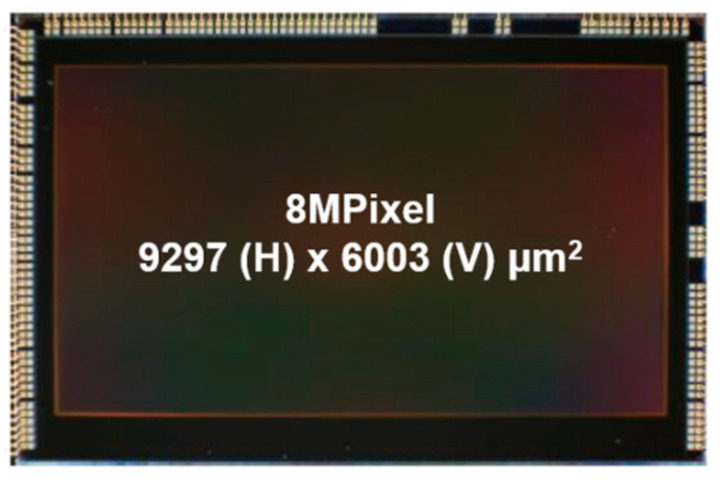
Chip micrograph of the 8Mp test chip.

**Figure 9 sensors-23-09150-f009:**
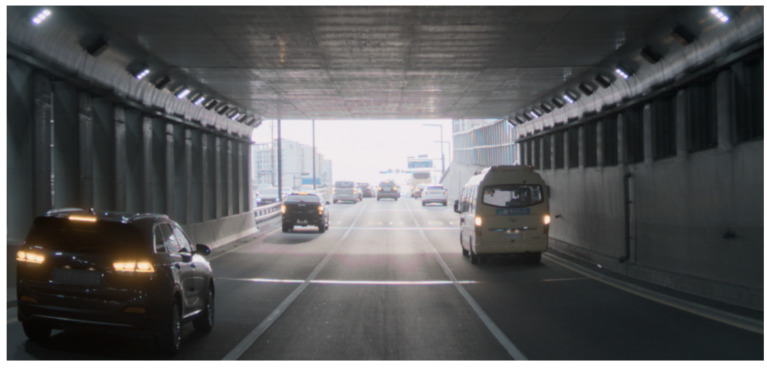
The HDR sample image of the 2.1 μm F-DTI pixel captured in tunnel using a full HDR mode with a single exposure.

**Figure 10 sensors-23-09150-f010:**
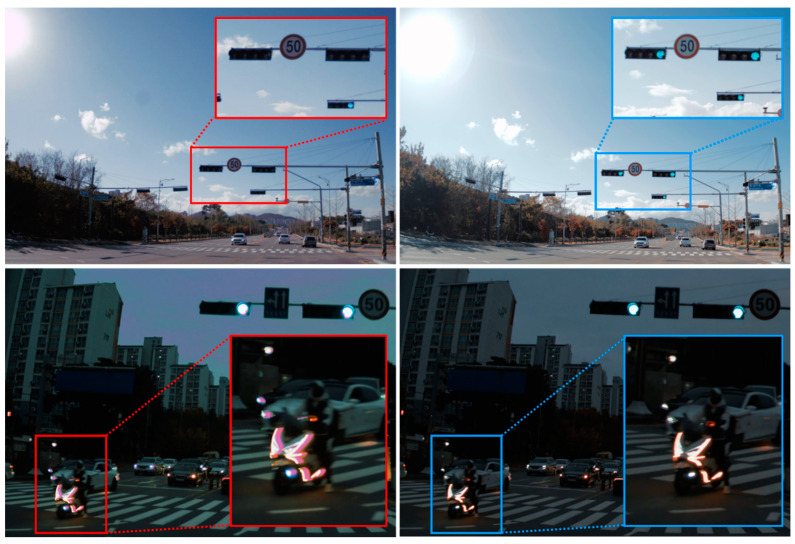
Outdoor sample images of the LED traffic lights (**up**) and the fast moving object (**down**) with and without the HDR mode. The upper right and lower right images show that there are no LED flicker and motion artifacts in HDR mode, respectively.

**Table 1 sensors-23-09150-t001:** Chip characteristics of the 2.1 μm F-DTI pixel.

Characteristics	Data
Process	Pixel 65 nm/Logic 28 nm, stacked BSI
Power supply	2.8 V/1.8 V/1.05 V
Si thickness	3.0 μm
Pixel array (H × V)	3840 × 2160
Max frame rate	36 fps @ 12 bit
FWC (LPD, SPD, in-pixel capacitor)	10 ke-, 10 ke-, 210 ke-
Sensitivity	32,000 e-/lux.sec (RCCB)
Sensitivity ratio	10
Conversion gain (HCG, LOFIC)	185 μV/e-, 4.2 μV/e-
Read noise (HCG)	0.83 e-
Single-exposure DR @ T_j_ 85 °C	120 dB
Minimum SNR dip @ T_j_ 105 °C	25 dB
Color filter array	RCCB (RGGB, RCCG, RYYCy support)

**Table 2 sensors-23-09150-t002:** HDR performance comparison.

HDR Characteristics	This Work	IEDM	IEDM	ISSCC	IISW
	(2.1 μm F-DTI)	2022 [7]	2021 [5]	2020 [6]	2019 [4]
Pixel pitch	2.1 μm	2.1 μm	2.1 μm	3.0 μm	3.0 μm
HDR technology	Sub-pixel	Sub-pixel	LOFIC	Sub-pixel	LOFIC
	LOFIC	LOFIC		LOFIC	
FWC (LPD, LOFIC)	10 ke-	10 ke-	10 ke-	12.8 ke-	20 ke-
	210 ke-	1.8 Me-	600 ke-	166 ke-	175 ke-
Single-exposure DR	120 dB	140 dB	110 dB	121 dB	96 dB
(based on total noises)	85 °C	85 °C	80 °C	85 °C	85 °C
Minimum SNR dip	25 dB	23 dB	25 dB	25 dB	25 dB
	105 °C	105 °C	125 °C	100 °C	100 °C

## Data Availability

Data are contained within the article.

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
