# Peer review of "Automotive 2.1 μm Full-Depth Deep Trench Isolation CMOS Image Sensor with a 120 dB Single-Exposure Dynamic Rangeâ€"

_sensors, 2023, doi:10.3390/s23229150_

Round 1
Reviewer 1 Report
Comments and Suggestions for Authors
Thank you for this very interesting paper submission. The results are impressive in such small pixel format.
I have a four suggestions to make the text more clear.
1) figure 3 and first paragraph of page 4: please explain in a bit more detail how the PSRR rejection scheme operates. How are the switches controlled which are showin in the PSRR compensation circuit in figure 3 (a)? Please add some text to explain this.
2) figure 4 and DSW switch in figure 2. The switch connects VDDA to VDDA. If this is the same net, there is no reason why this swich could accelerate discharging the in-pixel capacitor faster. Probably the left VDDA and right VDDA are not the same net (e.g. right VDDA is vertical line, left VDDA is horizontal line). Or do you rely on Cgs overlap capacitance of DSW? Please explain in more detail and use different labels if they are not the same net.
3) figure 2 : why is DRG pulsed during the last phase (SPD-LOF readout)? Please explain
4) suggestion for fig 5 to improve readability: indicate above X axis the readout mode that produces this SNR for the illumance level (from left to right: SPD-HCG, SPD-LCG, LPD-CDS, LPD-LOFIC). E.g. you could add some arrow bars with this text just above the X axis.
Reviewer 2 Report
Comments and Suggestions for Authors
This manuscript introduces a new developed high dynamic range CMOS sensor. The manuscript is quite clear and well-written. The test results presented are very useful for other groups working to develop CMOS sensors. I have no major issues with the paper, only a few minor items listed below, and once these are addressed I fully recommend the paper for publication.
Minor Issues:
1: What is the thickness of the epitaxial layer of this sensor? Please add this to Table 1;
2: The sensitivey ratio between large PD and small PD will cause PRNU in pixel level. What is the approximate value of PRNU? Does the HDR algorithm calibrate the pixel-level uniformity?
Reviewer 3 Report
Comments and Suggestions for Authors
DR of 120dB has been achieved based on sub-pixel structure and LOFIC. The most significant concern is the difference from the authors' IEDM2022 publication. In the IEDM2022 article, the pixel size is the same, and the DR is higher. That article is not mentioned in the introduction, which makes the advancement of the submitted paper ambiguous. The authors must cite their IEDM2022 publication and clarify why DR of 120dB is targeted while 140dB was already achieved.
Other comments and questions.
What is the dashed line in Figure 5.
What does DL mean in Figure 6(a)?
The authors should show the full spelling.
Line 184: What is the "estimated SNR curve"? How was it estimated based on what? Please clarify the model or calculation.
Line 194: The authors say, "we finally achieve the minimum SNR dip over 25dB at Tj 105..."
However, over 25dB is already obtained, as shown in Figure (b).
Why 25dB? If it is "around 30dB" or something, it is reasonable.
In Table 1, Minimum SNR dip@ Tj105 deg. is 25dB, which is not consistent with Figure 5(b) after curing.
HDR performance comparison is shown in Table 2. However, no discussion is given. Especially, the authors should explain how and how much the performance has been improved compared to IEDM2022[8].
Round 2
Reviewer 3 Report
Comments and Suggestions for Authors
The revision might be suitable for publication.